# Anodization of a Medical-Grade Ti-6Al-7Nb Alloy in a Ca(H_2_PO_2_)_2_-Hydroxyapatite Suspension

**DOI:** 10.3390/ma12183002

**Published:** 2019-09-16

**Authors:** Alicja Kazek-Kęsik, Izabela Kalemba-Rec, Wojciech Simka

**Affiliations:** 1Faculty of Chemistry, Silesian University of Technology, B. Krzywoustego Street 6, 44-100 Gliwice, Poland; 2Faculty of Metals Engineering and Industrial Computer Science, AGH University of Science and Technology, A. Mickiewicza Avenue 30, 30-059 Krakow, Poland; kalemba@agh.edu.pl

**Keywords:** titanium, plasma electrolytic oxidation, implant, hydroxyapatite

## Abstract

The electrochemical parameters used for surface treatments should be individually determined for each titanium alloy. In this paper, the parameters for the anodization of a medical-grade Ti-6Al-7Nb alloy in hydroxyapatite suspensions were determined. It was found that formation of a favorable porous oxide layer occurred for the plasma electrolytic oxidation process in a Ca(H_2_PO_2_)_2_ solution with 150 g/dm^3^ hydroxyapatite particles at 350 V and 450 V. The differences in the morphology, chemical and phase composition caused variability in the average surface roughness (up 4.25 μm) and contact angle (strongly hydrophilic) values. Incorporation of the hydroxyapatite ceramic particles into formed TiO_2_ layer also influenced the layer thickness and adhesion of the layers to the substrate. The oxide layers formed on the Ti-6Al-7Nb alloy were between 5.19 and 31.4 μm in thickness with an average range of approximately 8–15 μm. The formation of a ceramic layer under controlled electrochemical parameters allows the design of a bioactive surface of implants for bone tissue. The hydroxyapatite particles may promote the osseointegration process. Thus, in this study, the formation of ceramic composites on medical-grade Ti surfaces is presented and discussed.

## 1. Introduction

Electrochemical methods are often applied to modify the surface of metallic biomaterials [1]. Common commercial metallic biomaterials are made of stainless steel, cobalt-chromium alloys, titanium and selected Ti alloys (e.g., NiTi, Ti-6Al-4V, Ti-6Al-7Nb, and Ti-15Zr) [2,3]. There are new materials that may find applications in the medical field, especially as long-term implants attached to bone tissue. Titanium alloys composed of only nontoxic, biocompatible compounds often exhibit very good mechanical properties and have properties similar to those of bone. The mechanical properties of bone vary depending on the part of the bone being considered; thus, flexible materials and materials with a low Young modulus are desirable. The Ti-6Al-7Nb alloy is considered to be part of a new generation of materials that can be used as dental implants or orthopedic implants. It is a two-phase alloy, comprising alpha and beta phases, with an equilibrium state at room temperature that consists of the hexagonal phase stabilized with aluminum [4].

The surfaces of metallic biomaterials are treated using electropolishing and anodizing processes, including plasma electrolytic oxidation, electrophoretic deposition and electrochemical reduction [5,6,7]. The chemical and phase composition, as well as the morphology of the coatings, can be modified to obtain desirable features. The metallic substrate strongly influences the final physicochemical properties of coatings, even when the same electrochemical parameters are applied during the process. Depending on the implant application, the surface roughness (Ra) might be decreased as much as possible (e.g., for short-term implants), increased to 2 µm (e.g., for dental implants) or increased to 4 µm (e.g., for part of an endoprosthesis). The surface roughness, wettability and morphology of the material play a key role during the first stages of the material cytocompatibility with blood or selected cells (osteoblasts or fibroblasts) [8]. Electrochemical techniques can form ceramic layers, and the final effect of the surface treatment might be enhanced using post-treatment techniques, such as a heat or hydrothermal treatment, an alkali or acid treatment, or formation of additional layers using various techniques, including electrochemical techniques [9].

The plasma electrolytic oxidation process (PEO) is often used to obtain a well-adhering, compact layer with pores distributed at the layer top surface and in the middle part of the layer. The PEO process is carried out to incorporate chemical compounds that are soluble in water or ceramic particles from the anodizing bath [10]. The one step surface treatment reduces the material functionalization cost and avoids poor adhesion between interlayers, which may occur in multilayer materials. For biomedical applications, bioactive surfaces are recommended for long-term implants. To increase the bioactivity and biocompatibility of a material, a porous oxide layer may be enriched with various types of calcium phosphate particles, including hydroxyapatite, which is an inorganic component of human or animal bones. Particles of natural or synthetic hydroxyapatite exhibit osteoinductive properties and are often blended with hydrogels, incorporated into microparticles, and can also be deposited on the implants using electrophoretic techniques [11]. Ceramic particles are used for manufacturing ceramic scaffolds [12]. However, the mechanical properties of titanium implants support their long-term medical applications. The formation of an oxide layer enriched by hydroxyapatite particles on Ti implants could enhance the rapid osteointegration process.

In this study, the surface of a medical-grade Ti-6Al-7Nb alloy was modified by a one-step surface treatment using a plasma electrolytic oxidation process. Anodization was applied to obtain a porous oxide layer with incorporated bioactive particles of hydroxyapatite. The conditions of the anodizing process (voltage and composition of anodizing bath) were found to produce a bioactive surface, which may find application as long-term implants to bones.

## 2. Materials and Methods 

The Ti-6Al-7Nb alloy (BIMO, Poland) was purchased as a rod and cut into substrate samples 10 × 5 mm^2^ in size and 1 mm in thickness. Before anodization, the samples were ground using abrasive papers with grades up to 1200 grit. Then, samples were chemically etched in solution composed of 1 M HF and 4 M H_2_SO_4_ (time = 20 s) and cleaned in deionized water under sonication.

Then, the samples were anodized in three solutions composed of 0.1 M solution of Ca(H_2_PO_2_)_2_ with hydroxyapatite particles with concentrations of 50, 100 or 150 g/dm^3^. First, plasma electrolytic oxidation was carried out under galvanostatic conditions at 150 mA/cm^2^ and then under potentiostatic conditions with limiting voltages in the range of 150–450 V (DC power supply, PWR800H Kikusui, Japan equipped with Wavy software). After surface treatment, the samples were rinsed using deionized water and dried at room temperature.

The surface morphology and chemical composition were analyzed using scanning electron microscopy (SEM, Hitachi S-3400N, Tokyo, Japan, accelerating voltage = 25 kV) and energy-dispersive X-ray spectrometry (EDX, Thermo Noran, Thermo Scientific, Waltham, MA, USA). The thickness of the oxide layer formed on the substrate was determined according to the cross-section procedure presented in paper [13]. 

The surface roughness of the samples was determined by noncontact optical profilometry (Wyko N9300, Veeco, NY, USA). The arithmetical average roughness (Ra), root mean square roughness (Rq), maximum relative height (Rt) and maximum relative depth (Rz) were determined [13]. On the surface of each sample, 0.2 µL of high-purity deionized water was placed to determine the contact angle. The measurements were repeated 10 times at room temperature using DataPhysics (OCA 15EC, Filderstadt, Germany) equipment.

The phase composition of the coating layer was determined using powder X-ray diffraction (XRD) (Seifert 3003TT powder X-ray diffractometer (RICH. SEIFERT&CO. GmbH&Co. KG, Ahrensburg, Germany) with a copper X-ray tube, where kλ1 = 1.540598 Å, kλ2 = 1.544426 Å and kβ = 1.39225 Å; Ahrensburg, Germany). The XRD patterns were recorded over a 2θ range of 5° to 80° with a 0.05° step size.

The adhesion of the coatings was investigated using a Micro Scratch Tester (MST) from CSM Instruments applying a Rockwell C diamond indenter (serial number J-147, MA, USA, 100 μm radius) with loads from 30 mN–20 N, lengths of 5 mm and a table speed of 2.5 mm/min [14]. The results of the measurements are presented as an average of three measurements from three independent samples. 

## 3. Results

The morphology and chemical composition were determined for the samples anodized with various concentrations of hydroxyapatite particles (50, 100, and 150 g/dm^3^) with SEM and EDX, respectively. The EDX results as a function of voltage are shown in Figure 1.

The changes in the coating compositions depend on the voltage applied during the PEO process. Usually, when the voltage increases, the calcium content in the coatings also increases. The concentration of calcium and phosphorous incorporated into the coatings is a sum of the ions from the anodizing bath and incorporated hydroxyapatite ceramic particles. Thus, the Ca/P, Ca/Ti and P/Ti ratios varied. For the coating formed with a low concentration of hydroxyapatite (50 g/dm^3^), the calculated ratio Ca/P was high; however, the amount of incorporated phosphorous and calcium was low when the ratios were calculated with the substrate. These ratios increased when the process was carried out in the solution with an increased concentration of hydroxyapatite in the anodizing bath. Coatings formed in the solution composed of 150 g/dm^3^ hydroxyapatite particles comprised all elements with high ratios at the beginning of the anodization process. This means that coatings formed at 150 V in that solution were enriched with all bioactive compounds. For these coatings, the hydroxyapatite particles were also visible in the SEM images (see Figure 2). These layers were characterized in advance. Depending on the limiting voltages during the PEO process, the sample names were assigned as follows: TAN-200, TAN-250, TAN-300, TAN-350, TAN-400 and TAN-450 for 200, 250, 300, 350, 400 and 450 V applied voltages, respectively, during the PEO process.

The SEM images of the selected samples formed in solution containing the highest concentration of the bioactive compound are presented in Figure 2.

The surface morphology of the oxide layer strongly depended on the applied voltage during the PEO process. The characteristic microstructure of the oxide was observed when the voltage during the anodizing process was higher than 150 V. The surface of the TAN-200 sample is presented in Figure 2A. On the coating top, agglomeration of the hydroxyapatite particles was observed, and the agglomeration increased when the PEO process voltage increased to 250 V (TAN-250 sample, Figure 2B). The porous structure was clearly observed when the anodization voltage reached 300 V (TAN-300; Figure 2C), and the coating completely covered the titanium alloy surface. The pores were irregular with various shapes and sizes. The surface of the TAN-350 sample was also porous, and the pores were irregular with various shapes and sizes (Figure 2D). However, the structure of the coatings on the top of the layers was more reproducible than the surface observed for the TAN-300 sample. When the voltage increased to 400 V (Figure 2E), the coatings top melted, and the number of pores decreased in the top part of the layer. The pores were mostly round and were present throughout the top oxide layer. For the TAN-400 and TAN-450 samples, the agglomeration of incorporated hydroxyapatite particles was visible. However, the pores in the coating formed at 450 V (TAN-450; Figure 2F) were flakey and irregular. The significant changes in the surface morphology and chemical composition were observed for the titanium alloy samples anodized at 350 V and 450 V. Thus, for these surfaces, additional physicochemical analysis was performed. The limiting voltages applied during the PEO process also influenced the oxide layer thickness. The SEM cross sections and EDX mapping of the layers were obtained and they are shown in Figure 3 and Figure 4.

Voltage applied during the PEO process caused various layer thicknesses to form. For the TAN-350 sample, the average coating thickness was between 5.19 μm and 9.09 μm. The layer on the TAN-450 sample was thicker than that for the TAN-350 sample and was between 16.6 μm and 33.7 μm. Additionaly, EDX mapping of the oxide layers formed on TAN substrate was performed. Titanium, oxygen, calcium and phosphorous were detected in the oxide layers. Titanium is partially visible in the oxide layer, which is obvious due to formation of TiO_2_ during the anodizing process. EDX analysis confirmed that the calcium and phosphorous compounds were incorporated into coatings during the PEO process.

The surface roughness results determined by noncontact optical profilometry are presented in Figure 5. 

The surfaces of the Ti-6Al-7Nb alloy changed after the anodization process. The surface roughness increased for the coatings formed at 350 V and at 450 V. The difference between the samples is significant. For TAN-350, the parameters of surface roughness were as follows: Ra was 2.66 μm, Rq was 3.28 μm, and Rt was 59.68 μm. For TAN-450, Ra was 4.25 μm, Rq was 5.46 μm and Rt was 257.26 μm. The surface roughness and chemical composition influenced the coating wettability. The results of the contact angle measurements are given in Figure 6.

The formation of porous coatings caused the contact angle to decrease from 51.2° ± 4.7° to 36.5° ± 4.7°. Increasing the voltage during the anodization process caused the surfaces to become super hydrophilic, and the contact angle could not be determined.

Adhesion of the coatings formed on biomedical TAN alloy was examined using the scratch testing method with a Rockwell indenter. The difference in the adhesion of the coatings is clearly seen in Figure 7. For the TAN-350 sample, five stages of coating delamination at various friction forces were observed: Lc1 was 3.12 N ± 1.84, Lc2 was 7.33 ± 4.26 N, Lc3 was 13.55 ± 2.55 N, Lc4 was 19.79 ± 4.08 N, and Lc5 was 27.83 ± 0.93 N. The first and the last stages of the scratch-testing measurements are respectively presented in Figure 7A,B. At the beginning and the end of the scratch test, no cracks in the coatings were observed. The coatings on the TAN-350 sample were more continuous than the coatings on the TAN-450 sample. Four stages of coating delamination were observed for the TAN alloy anodized at an increased voltage (450 V): Lc1 was 4.35 ± 0.45 N, Lc2 was 8.51 ± 4.59 N, Lc3 was 12.60 ± 0.91 N, and Lc4 was 12.72 ± 0.16 N. The differences are also clearly observed in the optical micrographs shown in Figure 7C,D. At the beginning, the changes in the coating were due to rapid delamination and changes in the interface between the indenter and coatings (marked by red arrows). The coatings were continuous at the bottom layers, and the indenter penetrated the coatings rapidly, changing the coating structure near the bottom (Figure 7D, red arrows).

The phase composition of the samples was analyzed by XRD, and the patterns are presented in Figure 8.

The Bragg peaks coming from the substrate (Ti-6Al-7Nb alloy) decreased when the thickness of the coatings increased. For the TAN-350 sample, the shapes of the peaks recorded for the substrate were extended on the right side, which may indicate that nonstoichiometric oxides were formed. No signals from the hydroxyapatite particles or other calcium phosphate compounds were found. In the XRD pattern of the TAN-350 sample (Figure 8B), the bump between the two angles 5° and 20° 2θ indicates that an amorphous phase was formed during the anodization process. This result was also due to spark discharges that occurred during the PEO process because some areas of the coatings were melted. For the TAN-350 sample the calcium and phorous compounds were incorpoated into the oxide layer (see Figure 3), but they were in amorphous form. The characteristic bump is larger for the TAN-450 sample (Figure 8C) due to the formation of thicker coatings at higher voltage. For the TAN-450 sample, the TiO_2_ phase was determined (anatase), and small peaks from the hydroxyapatite particles were observed. Not all the signals were clearly resolved, especially in the low 2θ region of the XRD pattern due to the formation of the amorphous phase. However, the hydroxyapatite incorporated into the oxide layer was clearly detected as grains in the SEM images. The hydroxyapatite particles may have melted during the PEO process; thus, the signal from the amorphous material increased. If the hydroxyapatite particles melted, there may have been well-defined crystals still inside.

## 4. Discussion

Biomaterials made by titanium alloys are widely used as long-term implants. The formation of bioactive coatings should be designed for specific material applications. For example, dental implants should be characterized by a low surface roughness compared with implants for hip replacement. In this study, we analyzed the possibility of incorporating hydroxyapatite particles into a porous oxide layer at various voltages and using an anodizing bath with a concentration of hydroxyapatite particles between 50–150 g/dm^3^. The chemical compositions given in Figure 1 are presented as a ratio of the bioactive compounds to the alloying elements and phosphorous and showed different levels of calcium and phosphorous incorporation. The EDX analysis showed the results as an average of the compound that was soluble in the anodizing bath and elements from the hydroxyapatite particles as well. The changes in the chemical composition of the coatings, in terms of the amounts of both calcium and phosphorous, were observed when the PEO process was carried out mainly in the solution with 150 g/dm^3^ of hydroxyapatite.

In the SEM images, the particles of incorporated hydroxyapatite were also observed on the top of layers formed in this solution. The hydroxyapatite concentration also changed with the voltages that were applied during the PEO process, which induced changes in the layer thickness. These correlations are well known and have been reported in the literature [15]. However, it is not obvious that all of the coating would adhere well to the substrate and not be brittle. The differences between the layers formed at 350 V and 450 V were observed in their cross-sections (Figure 3 and Figure 4) and during the scratch testing. The coating formed at 450 V was not continuous and was more brittle than the coating formed at 350 V. During plasma electrolytic oxidation, spark discharges with different energies occur. The spark discharges occurred at the initial stage of the process, or sparks that are concentrated influence the formation of the layer. The final effects are related to the substrate and anodizing bath. However, for the TAN-450 sample, the voltages applied during the PEO caused the spark discharges to be more concentrated and the oxide layer to be less uniform and more porous than the oxide layer on the TAN-350 sample. In Figure 7A,C the differences during the first stage of nanoindentation and scratching of the coating are seen. The substrate for the TAN-350 sample is less visible than that for the TAN-450 sample. It was observed that, when using the same force during the measurement of both samples, it was easy to indent the TAN-450 ceramic layer. At the interface between the nanoindenter and the layer, delamination of the coating was observed for the TAN-450 sample, and the delamination continued until the end of the measurement. For the TAN-350 sample, no delamination or cracks were observed in the coatings. For both layers, no acoustic signals were registered, which means that neither of the coatings cracked during the scratch test. The values determined for the layers during the scratch test also confirmed that adhesion of the TAN-350 sample was much better than that for the TAN-450 sample. A higher force was applied to the TAN-350 sample than that for the TAN-450 sample to completely reveal the substrate. The scratch depth profiles registered for both samples (not shown here) were irregular. This was due to the porous structure of the layers and the incorporation of hydroxyapatite grains into the layers. Incorporation of the ceramic particles into the porous layers caused a slight difference in the scratch testing results, even when the same samples—but different areas of the samples—were used for the analysis. The SEM images indicate that particles agglomeration occurred at the top of the coatings. In the middle of the coatings, the distribution of the hydroxyapatite particles was variable (see cross-section).

In addition to the distribution of the hydroxyapatite particles, the voltage influenced the plasma process and some of the ceramic particles, and the oxides may have melted. The XRD patterns showed that the phase composition and the formation of crystals in the coatings depended on the electrochemical parameters. The particles incorporated into the layer at the high voltage melted and formed an amorphous-crystalline phase. However, some of the oxides may have also melted, especially at the top of the layer; thus, the characteristic bump coming from the amorphous phase was observed. The broad shape of the titanium alloy diffraction peaks may indicate the formation of nonstoichiometric oxides. This result is likely related to a thin, compact layer formed near the substrate. However, the titanium alloy used in these experiments was composed of titanium, niobium and aluminum. Niobium and aluminum form oxides very readily on their surfaces. These alloying elements affect the formation of the oxide layer, and as a result of the surface treatment, there was a mixture of TiO_2_ and Al and Nb oxides, even though they were not detected in the XRD patterns. In our previous paper, we confirmed that the alloying elements of anodized Ti-6Al-7Nb exist in the oxides. There was a small amount of these oxides; however, valve metals, like aluminum and niobium, readily form oxides [16].

The biocompatibility of implants is related to their wettability and surface roughness. A high surface roughness may inhibit adhesion of bone cells because they do not recognize the material and there are issues with the formation of an extracellular matrix [17]. Hydroxyapatite is considered a very bioactive material for long-term implants [18]. Formation of the hydroxyapatite during the anodization process also improves the corrosion resistance of the nano-structure titanium [19]. The application of hydroxyapatite particles might be limited due to the type of techniques available for ceramic particle deposition (e.g., plasma spraying, sol-gel or electrophoretic deposition) and the final effects of the mechanical properties of the coating, such as the adhesion of the particles to the substrate. The advantage of plasma electrolytic oxidation is the incorporation of hydroxyapatite particles that do not cover or fill the porous layer but induce adhesion of the osteoblast cells. It was reported that hydroxyapatite may be easily used for the formation of ceramic coatings, which will show very good adhesion to the metallic substrate [20]. The porous oxide layer is favorable for osteoblast cells when there is a variety of pores at the submicron and micron-scale [21]. Plasma electrolytic oxidation is a promising technique for obtaining ceramic layers with hydroxyapatite particles with desirable biological and mechanical properties. The addition of specific chemical compounds, in this case hydroxyapatite particles, may enhance the biocompatibility and bioactivity of titanium implant surfaces [22].

## 5. Conclusions

The medical-grade titanium alloy Ti-6Al-7Nb is able to be treated using the plasma electrolytic oxidation process with a solution composed of ceramic hydroxyapatite particles. The experiment has shown that desirable effects of the surface treatment are achieved when the PEO process is carried out at 350–450 V and in an anodizing bath with 150 g/dm^3^ of hydroxyapatite. Agglomeration of the hydroxyapatite on top of the porous oxide layer increased when the anodization occurred at 450 V. However, the oxide layer thickness then increased and became brittle, and the adhesion of the layer to the substrate substantially decreased. Anodization of titanium alloy at 450V caused the formation of amorphous phase, probably due to melting of the oxide layer and/or ceramic hydroxyapatite particles. The surface also became strongly hydrophilic. It seems that the layer formed at 350 V, is more promising for functionalization of Ti-6Al-7Nb alloy surfaces.

## Figures and Tables

**Figure 1 materials-12-03002-f001:**
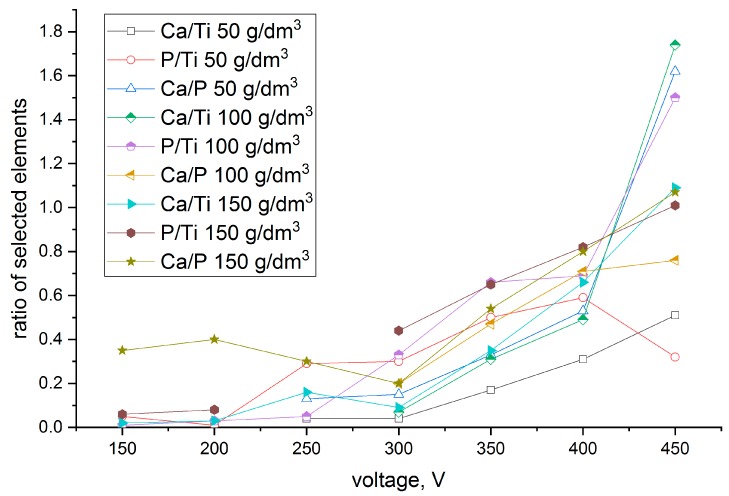
Constituent element ratios as a function of the voltage applied during the plasma electrolytic oxidation (PEO) process and concentration of hydroxyapatite in the anodizing bath. The results are presented as an average of the element content determined from the measured surface area (measurement repeated three times).

**Figure 2 materials-12-03002-f002:**
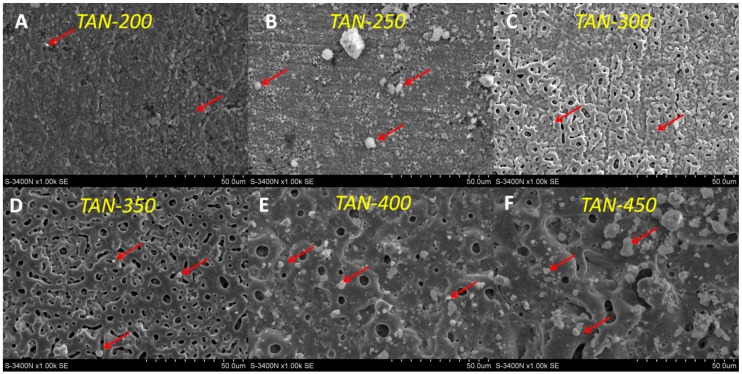
SEM images of the coatings formed on the Ti-6Al-7Nb surface in the solution composed of 150 g/dm^3^ hydroxyapatite particles at: (**A**) 200 V, (**B**) 250 V, (**C**) 300 V, (**D**) 350 V, (**E**) 400 V and (**F**) 450 V. Particles of hydroxyapatite are marked by red arrows.

**Figure 3 materials-12-03002-f003:**
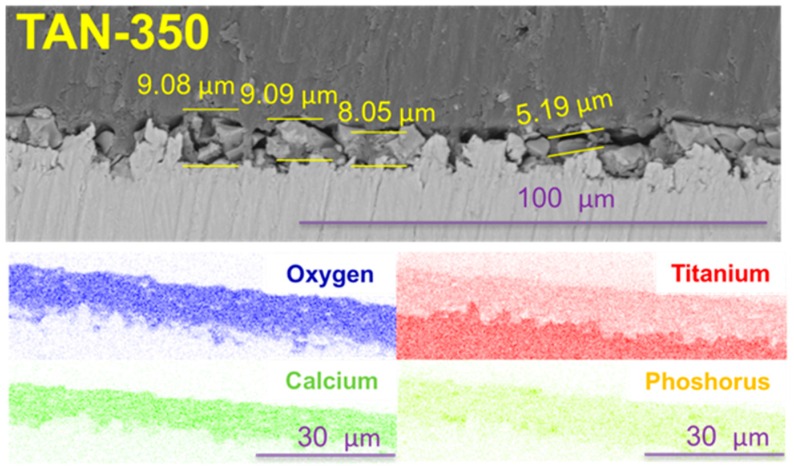
Cross sections of: TAN-350 and EDX mapping of selected part of oxide layer.

**Figure 4 materials-12-03002-f004:**
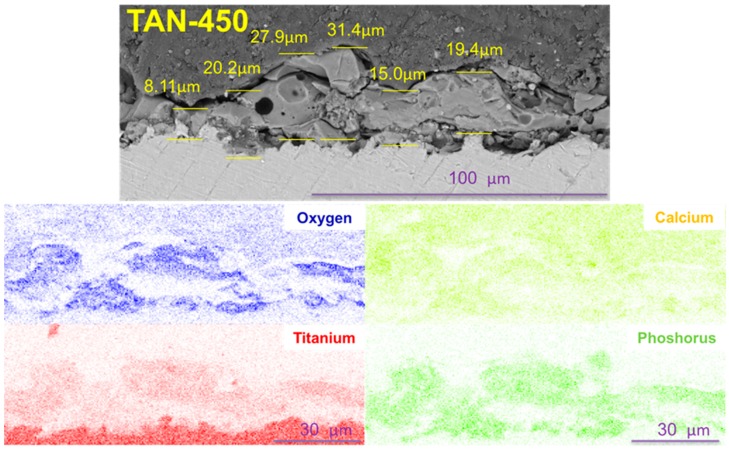
Cross sections of: TAN-450 and EDX mapping of selected part of oxide layer.

**Figure 5 materials-12-03002-f005:**
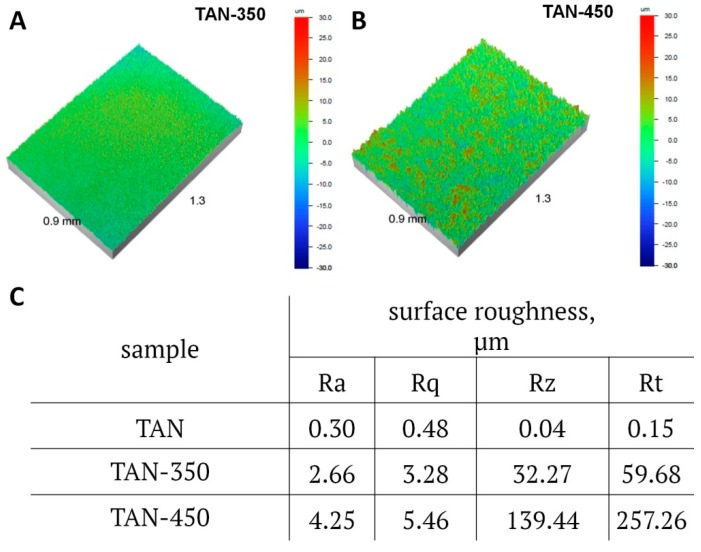
(**A**,**B**) Images of surface roughness of the samples measured from a selected area of the coatings (0.9 mm × 1.3 mm; magnification 5×), and (**C**) the results of the average measurements for each sample over selected areas of the coating surfaces.

**Figure 6 materials-12-03002-f006:**
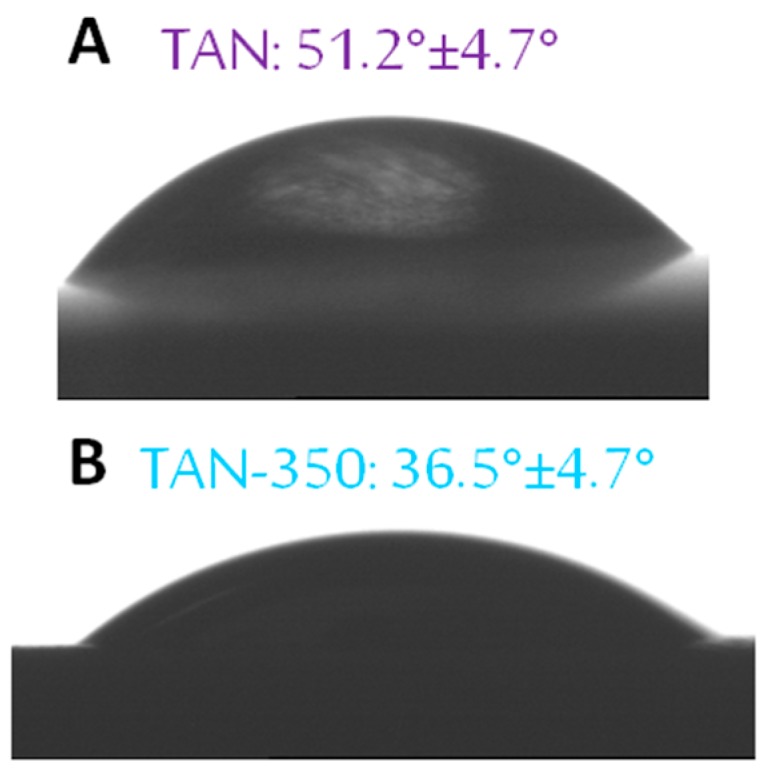
Contact angles of (**A**): TAN (**B**): TAN-350 samples

**Figure 7 materials-12-03002-f007:**
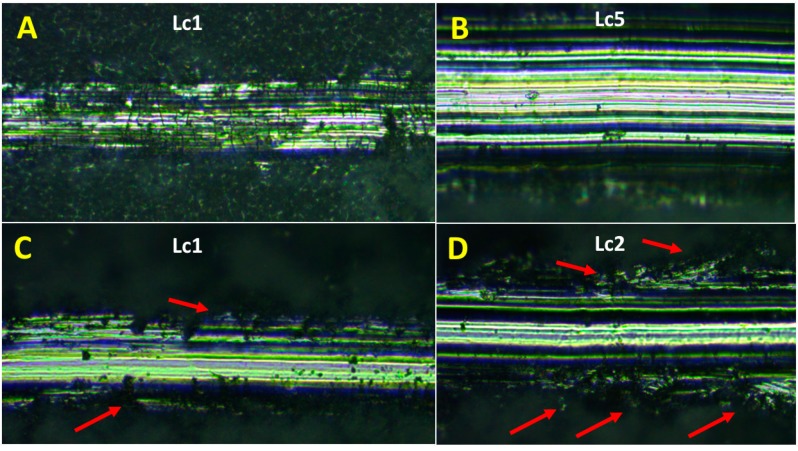
Optical micrographs of the (**A**,**B**) TAN-350 and (**C**,**D**) TAN-450 samples after the scratch testing.

**Figure 8 materials-12-03002-f008:**
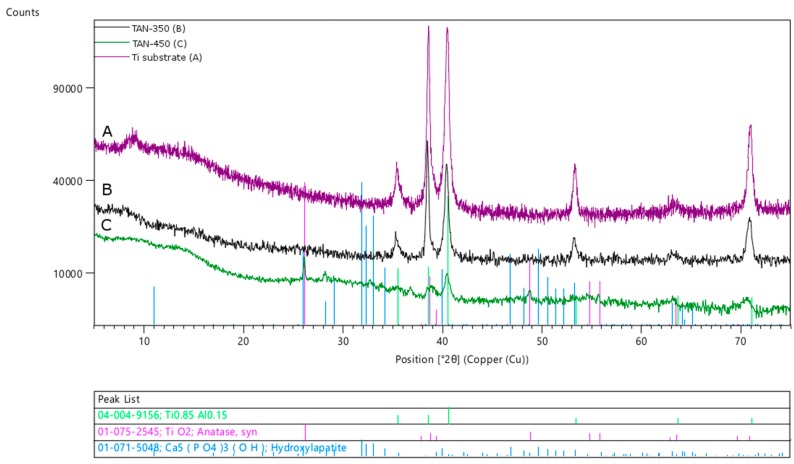
XRD pattern of: (**A**) TAN-350, (**B**) TAN-450 and (**C**) Ti substrate.

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
