# Peer review of "Anodization of a Medical-Grade Ti-6Al-7Nb Alloy in a Ca(H2PO2)2-Hydroxyapatite Suspension"

_materials, 2019, doi:10.3390/ma12183002_

Round 1
Reviewer 1 Report
My review is in attached file.

Author Response
Dear Reviewer, I would like to thank you for your time and valuable comments and suggestions. Manuscript was corrected according to your comments. Before manuscript submission, the English language was corrected by American Journal Experts (cerficiate was attached)
Please find below answers to your questions:
Page 2
The thickness of the samples was determined according to the cross-section procedure presented in paper [13].
However, the sample thickness was reported as being equal to 1 mm.
Thank you for the comment, the sentence was corrected as follow:
“The thickness of the oxide layer formed on the substrate was determined according to the cross-section procedure presented in paper [13].”
What is the reason to identify these particles as hydroxyapatite? Page 4
The differences in surface morphology indicate that the ceramic particles from suspensions were incorporated into the oxide layer. In our previous work, we carried out the PEO process in Ca(H2PO2)2 solution, and we didn’t observe formation of ceramic particles. On the XRD patterns we also did not observe ay signals coming from another calcium phosphates compounds.
The broad shape of the titanium diffraction peaks may indicate the formation of nonstoichiometric oxides.
However, titanium metal was not observed by the XRD analysis. This sentence should be corrected.
Thank you for your comment, we have analyzed the titanium alloy as a substrate. The sentence in the manuscript was corrected.

Reviewer 2 Report
Review of paper no. 581094 titled Anodization of a medical-grade Ti-6Al-7Nb alloy in a hydroxyapatite suspension by A. Kazek-Kesik et al.
The authors studied the anodizing of a Ti alloy by hydroxyapatite for medical applications. A plasma electrolytic oxidation process was used to obtain adherent and dense layers. The paper brings in some interesting results for the community. On the hand, the paper needs to improve discussion before being accepted for publication. I recommend a major revision in line with the following comments:
1.It is written in the materials and methods section (lines 78-79) that samples were anodized in a 0.1 M solution of hydroxyapatite. However, in the same sentence it is stated that particles with concentrations of 50 g/dm3, 100 g/cm3 or 150 g/dm3 were used. One or three different solutions have been studied? As the molar mass of hydroxyapatite is approximately 500 g/mol, the 0.1 M solution corresponds to 50 g/dm3. Was it possible to dissolve such amount of hydroxyapatite in water? Please, explain how you prepared your solutions.
2.The formula given for hydroxyapatite (Ca(H2PO2)2, line 78) is wrong as it leads to a mono-valent phosphorus. The valency of P is 5+ and the correct formula is Ca5(PO4)3(OH).
3.The microstructure of the prepared layers (Fig. 2) should be presented before the chemical analysis. Furthermore, the hydroxyapatite particles should be arrow-marked on the same figure.
4.I do not understand why the Ca/Ti and P/Ti ratios have been measured (Fig. 1). Ca and P are parts of the coating, however; Ti comes from the substrate. Was the alloy somehow interacting the coating? If so, the element line scans (EDS) of the cross-section would be more helpful. These data should be provided to see the coating-substrate interactions, see also comment 7 below.
5.The EDS line scans can be combined with Fig. 3. The theoretical Ca/P ratio is 5/3=1.67. This value, however, has been obtained for the 50 g/dm3 solution only (Fig. 1). Please, explain why this ratio sharply dropped to 0.8 or 1.0 for the 100 g/dm3 and 150 g/dm3 solutions respectively.
6.I recommend presenting the actual images of the wetting experiments (droplets) instead of Fig. 5. The contact angles can be written below each droplet.
7.Fig. 7 indicates that there could have been a titania (TiO2) formed at the interface. This suggests a possible substrate oxidation. I recommend presenting either element line-scans or maps for each cross-section concerned. These data could help to see whether the other oxides (Al2O3, Nb oxides) have also been formed. The interactions at the coating-substrate interface need to be thoroughly discussed and compared. Furthermore, as this paper deals also with the surface structuring of the hydroxyapatite coatings, I recommend comparing the results with recent studies. The following papers should be referenced: Ceramics International 44 (2018) 1250–1268; RSC Adv., 2017, 7, 33285;
RSC Adv., 2016, 6, 72733.
End of comments
Author Response
I would like to thank you Reviewer for the comments and suggestions.
Before manuscript submission, the English language was corrected by American Journal Experts (cerficiate was attached).
Please find below answers to your questions.
It is written in the materials and methods section (lines 78-79) that samples were anodized in a 0.1 M solution of hydroxyapatite. However, in the same sentence it is stated that particles with concentrations of 50 g/dm3, 100 g/cm3 or 150 g/dm3 were used. One or three different solutions have been studied? As the molar mass of hydroxyapatite is approximately 500 g/mol, the 0.1 M solution corresponds to 50 g/dm3. Was it possible to dissolve such amount of hydroxyapatite in water? Please, explain how you prepared your solutions.
Thank you for your comments. PEO process was carried out in three different solutions, the description of the anodization process was corrected. The PEO process was carried out in Ca(H2PO2)2 solution with the hydroxyapatite particles (it was a suspension).
2.The formula given for hydroxyapatite (Ca(H2PO2)2, line 78) is wrong as it leads to a mono-valent phosphorus. The valency of P is 5+ and the correct formula is Ca5(PO4)3(OH).
Thank you for your comment, the PEO process was carried out in soluble in water (Ca(H2PO2)2 with Ca5(PO4)3(OH) particles.
3.The microstructure of the prepared layers (Fig. 2) should be presented before the chemical analysis. Furthermore, the hydroxyapatite particles should be arrow-marked on the same figure.
Thank you for your suggestion. We have decided to present the results of chemical composition of various coatings to determine the best treatment of the Ti-6Al-7Nb alloy. Then, we have presented the surface morphology. Because, we have a lot of the results (the SEM images), we thought that we should present the surface morphology of the selected coatings, which will analyzed in details.
4.I do not understand why the Ca/Ti and P/Ti ratios have been measured (Fig. 1). Ca and P are parts of the coating, however; Ti comes from the substrate. Was the alloy somehow interacting the coating? If so, the element line scans (EDS) of the cross-section would be more helpful. These data should be provided to see the coating-substrate interactions, see also comment 7 below.
5.The EDS line scans can be combined with Fig. 3. The theoretical Ca/P ratio is 5/3=1.67. This value, however, has been obtained for the 50 g/dm3 solution only (Fig. 1). Please, explain why this ratio sharply dropped to 0.8 or 1.0 for the 100 g/dm3 and 150 g/dm3 solutions respectively.
Formation of the oxide on the valve metals causes, that the mixture of the various oxides are formed on the substrate. Depends on the chemical composition of the anodizing bath, the oxides might be composed of the chemical compounds coming from anodizing bath. It depends on the electrochemical parameters of the PEO process and chemical composition of anodizing bath, as well. Thus, we also calculate ratio of the elements incorporated from anodizing bath (Ca, P), to the titanium substrate. There is no clear interface between oxide layer and substrate, due to the nature of anodization process. In our previous work (KrzÄ…kaÅ‚a, A.; et al., W. Formation of bioactive coatings on a Ti–6Al–7Nb alloy by plasma electrolytic oxidation. Electrochim. Acta. 2013, 104, 407-424.) we have presented the EDX mapping for the cross-section samples. The nature of the anodization process is well known, thus in my opinion is better to show how the ratio of these elements may change when we various anodizing bath. For all of the samples, the EDX line will be very similar, as for the anodized valve metals.
The difference in compounds ratio is due to concentration of the ceramic particles in the bath. Because we have suspensions, the spark discharges occur during the PEO process will be different. The amount of incorporated Ca and P compounds from Ca(H2PO2)2 (soluble in water) and then incorporation of ceramic particles is variety. We can control electrochemical parameters and composition of anodizing bath, however we cannot control the spark discharges, thus the results are different (strongly depends of the concentration of ceramic particles in the bath). In our case, it was important to increase concentration of bioactive compounds (hydroxyapatite particles) in the oxide layer. Thus, beside the Ca/P ratio, we calculate the Ca/Ti and P/Ti ratio. The most promising results were obtained for the samples anodized in solution with 150 g/L of hydroxyapatite, and these samples we analyzed in details.
6.I recommend presenting the actual images of the wetting experiments (droplets) instead of Fig. 5. The contact angles can be written below each droplet.
Thank you for your comment, the Fig. 5 was corrected.
7.Fig. 7 indicates that there could have been a titania (TiO2) formed at the interface. This suggests a possible substrate oxidation. I recommend presenting either element line-scans or maps for each cross-section concerned. These data could help to see whether the other oxides (Al2O3, Nb oxides) have also been formed. The interactions at the coating-substrate interface need to be thoroughly discussed and compared. Furthermore, as this paper deals also with the surface structuring of the hydroxyapatite coatings, I recommend comparing the results with recent studies. The following papers should be referenced: Ceramics International 44 (2018) 1250–1268; RSC Adv., 2017, 7, 33285;
RSC Adv., 2016, 6, 72733.
Thank you for the comments, the result of anodization is formation of titanium oxides, thus we did not present the EDX mapping or EDX line. In our case, we have the titanium oxide layer and incorporated the ceramic particles into the porous layer. It is not only hydroxyapatite layer, with the interface with oxides. Thus, we discussed that the Al2O3 or Nb2O5 oxide could be formed, but they were not detectable by XRD techniques. Due to the nature of anodization process of valve metals, these oxides were formed, and we discuss this phenomena in our manuscript. However, the last reference (RSC Adv., 2016, 6, 72733) is linked to the study about formation of nanocrystalline carbonated hydroxyapatite with polylactic acid, and I am not sure how to properly find the relation between that work and out manuscript. Thank you for your comment, I corrected the manuscript and improve discussion according to your suggestions.

Round 2
Reviewer 1 Report
After the revision, the paper is greatly improved and, in my opinion, it can be published in the present state.
Author Response
I would like to thank Reviewer for the comments and suggestions which caused that the manuscript is more readable.
Reviewer 2 Report
Review of paper no. 581094 titled Anodization of a medical-grade Ti-6Al-7Nb alloy in a hydroxyapatite suspension by A. Kazek-Kesik et al.
This is a revised version of a previously reviewed paper. Some comments have been answered. However, the paper still requires a major revision. Some formal changes have been made; however, a detailed characterization of the anodized layers is missing. The paper is very short as it lacks some important materials science results. This is a stand-alone paper. It is not sufficient to refer to previous papers for EDX analysis, it should be presented here. The authors have to resolve this issue; otherwise their paper cannot be accepted by a materials science journal. The following comments should be considered:
1.It has now been explained in the cover letter that a mixture of calcium hypophosphite (Ca(H2PO2)2) and hydroxyapatite was used during anodizing. As such, the title should be rewritten in the following manner: Anodization of a medical-grade Ti-6Al-7Nb alloy in a Ca(H2PO2)2-hydroxyapatite suspension.
2.The chemical analysis (Fig. 1) shows that the Ca/P ratio was less than 1 for samples anodized at 350 V. As such, it is clear that the hydroxyapatite layer (Ca/P ratio 1.67) has not been formed to a large extent. You have to explain which compound/compounds have been actually formed on the sample surface as a result of anodizing.
3.The layers are not sufficiently characterized in terms of their chemical composition and phase constitution. The XRD analysis provided in Fig. 7 does not help to reveal the product phases either. For the sample anodized at 350 V there are hardly any peaks visible except for the substrate. It is crucial to provide the EDX chemical analysis. The EDX line scans/maps should be combined with Fig. 3. If there was any interaction at the coating-substrate interface, it should be described.
4.It is not clear what was the upper layer in Fig. 3. Was it an epoxy resin? How were the cross-sections prepared?
5.Results in Fig. 1 indicate that the hydroxyapatite coating has been formed for the samples anodized at 450 V (Ca/P ratio ~ 1.6). Taking this result into consideration, why is it concluded that anodizing at 450 V was less promising (line 472)?
6.I recommend presenting only the droplets in Fig. 5 to save the journal’s space. The graph is not necessary. The contact angles can be written directly below each droplet.
End of comments
Author Response
I would like to thank Reviewer for the comments and suggestions which caused that the manuscript is more readable. Please find below our answers on your questions:
1.It has now been explained in the cover letter that a mixture of calcium hypophosphite (Ca(H2PO2)2) and hydroxyapatite was used during anodizing. As such, the title should be rewritten in the following manner: Anodization of a medical-grade Ti-6Al-7Nb alloy in a Ca(H2PO2)2-hydroxyapatite suspension.
Thank you for your suggestion, title of the manuscript was corrected.
The chemical analysis (Fig. 1) shows that the Ca/P ratio was less than 1 for samples anodized at 350 V. As such, it is clear that the hydroxyapatite layer (Ca/P ratio 1.67) has not been formed to a large extent. You have to explain which compound/compounds have been actually formed on the sample surface as a result of anodizing.
Thank you for your comment. According to the XRD analysis the coatings were mainly form of TiO2 (anatase),and particles of hydroxyapatite (visible on the SEM images, and recorded by XRD as well). The part of coating were analysed by EDX and the final ratio is sum of the elements incorporated into the layer from anodizing bath (soluble Ca(H2PO2)2 and HA particles. In the manuscript we explain it as follow:
" The changes in the coating compositions depend on the voltage applied during the PEO process. Usually, when the voltage increases, the calcium content in the coatings also increases. The concentration of calcium and phosphorous incorporated into the coatings is a sum of the ions from the anodizing bath and incorporated hydroxyapatite ceramic particles."
3.The layers are not sufficiently characterized in terms of their chemical composition and phase constitution. The XRD analysis provided in Fig. 7 does not help to reveal the product phases either. For the sample anodized at 350 V there are hardly any peaks visible except for the substrate. It is crucial to provide the EDX chemical analysis. The EDX line scans/maps should be combined with Fig. 3. If there was any interaction at the coating-substrate interface, it should be described.
Thank you for your comment. In our last response, I would like to show, that titanium partially exist in TiO2, and it is natural due to anodization process and formation the oxide layer, in this case titanium oxide layer. Thus, I wrote about the results in our previous paper, but only in response on review. According to your suggestions we analyzed the layer one more time, and the EDX mapping of cross-section of the oxide layers formed on substrate was performed. Please see the Figure 3, where we add the results. Description of the results was also corrected. All of the layer were also analyzed by EDX measurements, and the ratio of these elements are presented in the first part of our manuscript. The description about phase composition of TAN-350 sample was also corrected. The incorporated calcium and phosphorous compounds were in amorphous form, thus we observed the characteristic bump on XRD pattern.
Thank you for your comment, due to your suggestions the manuscript is much more readable.
4.It is not clear what was the upper layer in Fig. 3. Was it an epoxy resin? How were the cross-sections prepared?
It was a standard procedure for preparation of cross-section, where the epoxy resin was used. In materials in methods we wrote that the produce is present in our previous paper (page 2: The thickness of the oxide layer formed on the substrate was determined according to the cross-section procedure presented in paper [13]."
5.Results in Fig. 1 indicate that the hydroxyapatite coating has been formed for the samples anodized at 450 V (Ca/P ratio ~ 1.6). Taking this result into consideration, why is it concluded that anodizing at 450 V was less promising (line 472)?
Beside chemical composition of the coatings, mechanical adhesion of the coating to the substrate was measured. Based, on the results it was concluded that the much better adhesion exhibit coating formed at 350 V. On the other hand, not only chemical composition of the layer (based on Ca/P ratio) determine good biocompatibillity or bioactivity of the material. Because PEO layer is used to anodize real shape implants, the adhesion of coating to the implant is very importat. We correlated the results with the surface roughness and wetabillity, and we explain it as follow:
" Agglomeration of the HA on top of the porous oxide layer increased when the anodization occurred at 450 V. However, the oxide layer thickness then increased and became brittle, and the adhesion of the layer to the substrate substantially decreased. The oxide layer formation an increased voltage (450 V) caused an formation of amorphous phase, probably due to melting of the oxide layer and/or ceramic HA particles. The surface also became strongly hydrophilic. It seems that the layer formed at 350 V, is more promising for functionalization of Ti-6Al-7Nb alloy surfaces."
6.I recommend presenting only the droplets in Fig. 5 to save the journal’s space. The graph is not necessary. The contact angles can be written directly below each droplet
According to the suggestion, the figure was corrected.
Round 3
Reviewer 2 Report
Review of paper no. 581094 titled Anodization of a medical-grade Ti-6Al-7Nb alloy in a Ca(H2PO2)2-hydroxyapatite suspension by A. Kazek-Kesik et al.
This is a revised version of a previously reviewed paper. Most of the comments have been sufficiently answered. The paper is now acceptable for publication subject to minor revision. The following points should be considered:
1.It should be clearly stated in the abstract itself that the anodized layer was composed of TiO2 and Ca(H2PO2)2-hydroxyapatite particles.
2.If the Ca(H2PO2)2/hydroxyapatite particles can be distinguished (line 147 in the manuscript), they should be clearly labelled (arrow-marked) in Fig. 2.
3.As the upper layer (black layer) in Figs. 3a and 3b is epoxy, it needs not to be presented. You can skip these two images as most information is already provided in Figs. 3c and 3d.
4.Details of the anodized layer (Figs. 3c and 3d) are sufficient. However, the EDS maps provided are unreadable. They should be presented individually. Please, provide a separate map for each element (O, Ti, P, Ca, Al, Nb) in Fig. 3.
End of comments
Author Response
Dear Reviewer, the manuscript was corrected according to your suggestions. Thank you for all your comments.
1.It should be clearly stated in the abstract itself that the anodized layer was composed of TiO2 and Ca(H2PO2)2-hydroxyapatite particles.
Abstract was corrected.
2.If the Ca(H2PO2)2/hydroxyapatite particles can be distinguished (line 147 in the manuscript), they should be clearly labelled (arrow-marked) in Fig. 2.
Particles of hydroxyapatite were marked by red arrows.
3.As the upper layer (black layer) in Figs. 3a and 3b is epoxy, it needs not to be presented. You can skip these two images as most information is already provided in Figs. 3c and 3d.
4.Details of the anodized layer (Figs. 3c and 3d) are sufficient. However, the EDS maps provided are unreadable. They should be presented individually. Please, provide a separate map for each element (O, Ti, P, Ca, Al, Nb) in Fig. 3
Thank you for your suggestion, Fig. 3 was corrected.